# Integrating multiple spatial transcriptomics data using community-enhanced graph contrastive learning

**Wenqian Tu, Lihua Zhang** ⓘ *

School of Artificial Intelligence, School of Computer Science, Wuhan University, Wuhan, China

* zhanglh@whu.edu.cn

## Abstract

Due to the rapid development of spatial sequencing technologies, large amounts of spatial transcriptomic datasets have been generated across various technological platforms or different biological conditions (e.g., control vs. treatment). Spatial tran-scriptomics data coming from different platforms usually has different resolutions. Moreover, current methods do not consider the heterogeneity of spatial structures within and across slices when modeling spatial transcriptomics data with graph-based methods. In this study, we propose a community-enhanced graph contrastive learning-based method named Tacos to integrate multiple spatial transcriptomics data. We applied Tacos to several real datasets coming from different platforms under different scenarios. Systematic benchmark analyses demonstrate Tacos's superior performance in integrating different slices. Furthermore, Tacos can accurately denoise the spatially resolved transcriptomics data.

## Author summary

Integrating multiple spatial transcriptomics datasets can provide more comprehensive understanding of tissue environment. However, batch effects and different sequencing resolutions make the integrative analysis of various spatial datasets be challenge. Current integration methods require spatial transcriptomics data with similar structures and similar resolutions, which might be violated for real heterogeneous datasets. We present Tacos, which adopts a community-enhanced contrastive graph neural network method to model the spatial transcriptomics data by considering heterogenous structures. Tacos uses a triplet loss to facilitate the alignment of different slices by pulling the mutual nearest neighbor pairs between spots from different slices close and pushing the randomly selected negative pairs away. Applications on spatial transcriptomics coming from various sequencing platforms demonstrate that Tacos is not only efficiently remove batch effects but also preserve the biological structures, especially on the spatial transcriptomics data with different resolutions. Moreover, Tacos is able to denoise the spatial transcriptomics data.

**Data availability statement:** Source codes and tutorials have been deposited at the GitHub repository (https://github.com/0617spec/tacos; https://anonymous.4open.science/r/tacos-5EDE).

**Funding:** The study is supported by the National Natural Science Foundation of China (Grant Number: 62202343 to LZ) and the Key Technologies Research and Development Program (Grant Number: 2023YFF0725400 to LZ). The funders had no role in study design, data collection and analysis, decision to publish, or preparation of the manuscript.

**Competing interests:** The authors declare that they have no competing interests.

## Introduction

Spatial transcriptomics (ST) sequencing technologies provide both spatial location information and gene expression information, enabling comprehensive characterization of gene expression patterns in the context of tissue microenvironment [1]. It's well known that the biological functions are highly associated with spatial information. ST technologies have been wildly adopted to study neuroscience, plant biology and disease research [2,3]. An increasing number of spatial transcriptomic datasets have been generated with varying spatial resolutions. For example, 10x Visium achieves a spatial resolution of 55 μm [4], Slide-seq with a spatial resolution of 10 μm approaches single-cell resolution [5,6], Stereo-seq and seqFISH achieve subcellular resolution [7,8]. However, these ST datasets often suffer from severe noise as the shallow nature of sequencing for each spot and other steps in preserving the spatial locations of sequencing.

Many data integration methods for single cell transcriptomics data have been proposed [9] (i.e., Seurat [10], Harmony [11] and scMC [12]), while they cannot be applied spatial transcriptomics data as the spatial coordinates are not considered. Some computational methods have been proposed for processing and analyzing spatial transcriptomics data. For example, SpaGCN [13] incorporates gene expression and spatial coordinates with extra histological images information to detect spatial domains by graph convolutional network. SpaceFlow [14] utilizes spatially regularized graph contrastive neural network to integrate spatial information and gene expression data, generating low-dimensional embeddings. STAGATE [15] adopts a graph attention autoencoder framework to selectively integrate information from neighboring spots. GraphST [16] combines graph contrastive neural network and self-supervised learning to fully leverage spatial information and gene expression profiles for various analytical tasks. These methods have been comprehensively benchmarked in a previous study [17]. However, SpaGCN, SpaceFlow and STAGAGT are designed for single slice, which cannot be applied to integrating multiple spatial transcriptomics data.

Slices coming from different experimental conditions or technology platforms or section locations often exhibit differences in various levels. Several computational methods have been proposed to integrate multiple slices of spatial transcriptomics data. PASTE [18] applies optimal transport algorithm to align different slices. GPSA [19] uses deep Gaussian process to align the spatial coordinates of different slices. These methods perform well in integrating multiple slices at the same resolution. STAligner [20] and SLAT [21] integrate multiple slices based on graph neural network. SPIRAL align multiple slices by combining graph neural network and optimal transport method [22]. They do not consider the different community density between slices from different resolutions, which may limit their performance in integrating multiple slices coming from different platforms.

To fill this gap, we developed a multiple spatial **Tra**nscriptomics data integration method using **co**mmunity-enhanced graph contra**s**tive learning (Tacos). We applied Tacos to several real datasets, including multiple human cortex slices with the same platform, two mouse olfactory bulb slices from different platforms, two mouse embryo slices with different spatial resolutions, and spatial transcriptomic data of healthy and Alzheimer coming from different platforms. Systematic benchmark analyses with other methods show that Tacos is an efficient method for integrating multiple spatial transcriptomic datasets across different conditions.

## Results

### Overview of Tacos

Tacos takes both the spatial coordinates and the normalized gene expression profiles as inputs. As depicted in Fig 1, Tacos first constructs spatial graph for each slice based on its spatial

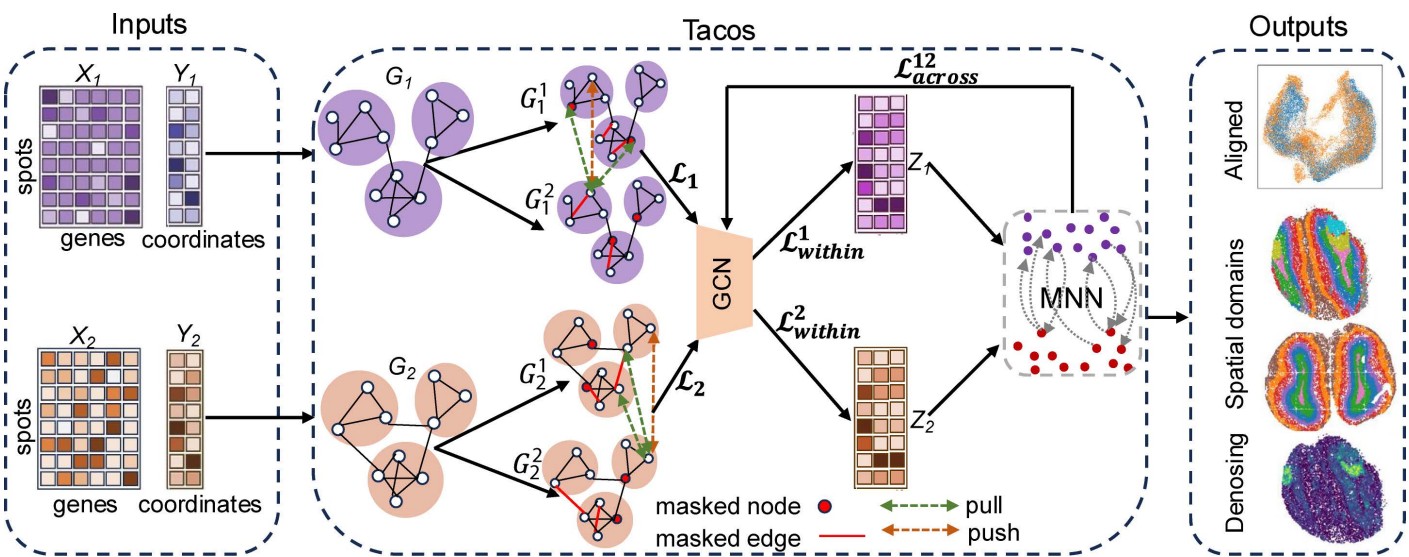

**Fig 1. Overview of Tacos.** Here, we take two spatial transcriptomics data as an example. Tacos treats these two normalized gene expression matrices $X_1$, $X_2$ and their corresponding spatial coordinates $Y_1$, $Y_2$ as inputs. Tacos builds graphs $G_1$ and $G_2$ based on $Y_1$ and $Y_2$. Two augmented graph views are generated for $G_1$ and $G_2$. A two-layer GCN encoder is adopted to extract embeddings $Z_1$ and $Z_2$. To align the slices, we identified MNN pairs based on $Z_1$ and $Z_2$. Tacos adopts a triplet loss to pull the MNN pairs close and push the negative pairs away to align the embeddings, where the negative pair are two spots with one spot from MNN and another spot being randomly selected. The total loss of Tacos is composed with three different constraints (contrastive constraint $\mathcal{L}_{t,t=1,2}$, within-slice constraint $\mathcal{L}_{within}^{t,t=1,2}$ and across-slice constraint $\mathcal{L}_{across}^{12}$). Red edges and red nodes are masked ones.

coordinates. Then a graph contrastive learning-based encoder is used to extract spatially aware embedding of each slice. Considering heterogenous spatial structures within each slice or across different slices, Tacos adopts communal attribute voting and communal edge dropping strategies to generate augmented graph views (Methods). Specially, communal attribute voting strategy detects nodes' features that are more likely to be masked. And communal edge dropping strategy is used to compute edges' mask probabilities. With the community-enhanced contrastive learning-based encoder, Tacos obtains the spatially aware embeddings. Next, Tacos detects mutual nearest neighbor (MNN) pairs between spots from different slices based on the spatially aware embeddings to facilitate the alignment of different slices. Tacos treats the MNN pairs as positive pairs and treats randomly selected spots as negative points. Then Tacos adopts a triplet loss to pull the positive pairs close and push the negative pairs away to update the embeddings. Finally, downstream analyses can be done on the embeddings, which are used for denoising the spatial transcriptomics data (Methods: "Spatial Transcriptomics Data Denoising").

## Tacos achieves superior alignment performance on different slices from the same platform

We applied Tacos to the ST datasets of the dorsolateral prefrontal cortex (DLPFC) from 10x Visium. Firstly, we focused on the adjacent slices (slice number: 151674 and 151675) from the same donor. We compared Tacos with Scanpy [23], Harmony [11], SLAT [21], SPIRAL [22] and STAligner [20]. Tacos, SLAT, Harmony, SPIRAL and STAligner could remove the batch effect which can be visualized in the UMAP [24] space (Fig 2A). Then we inferred the developmental trajectory using PAGA [25] on the integrated embeddings. PAGA estimates the connectivity of manifold partitions, providing an interpretable, graph-like map of the manifold of embedding. A good integrated embedding should preserve the linear structure of different

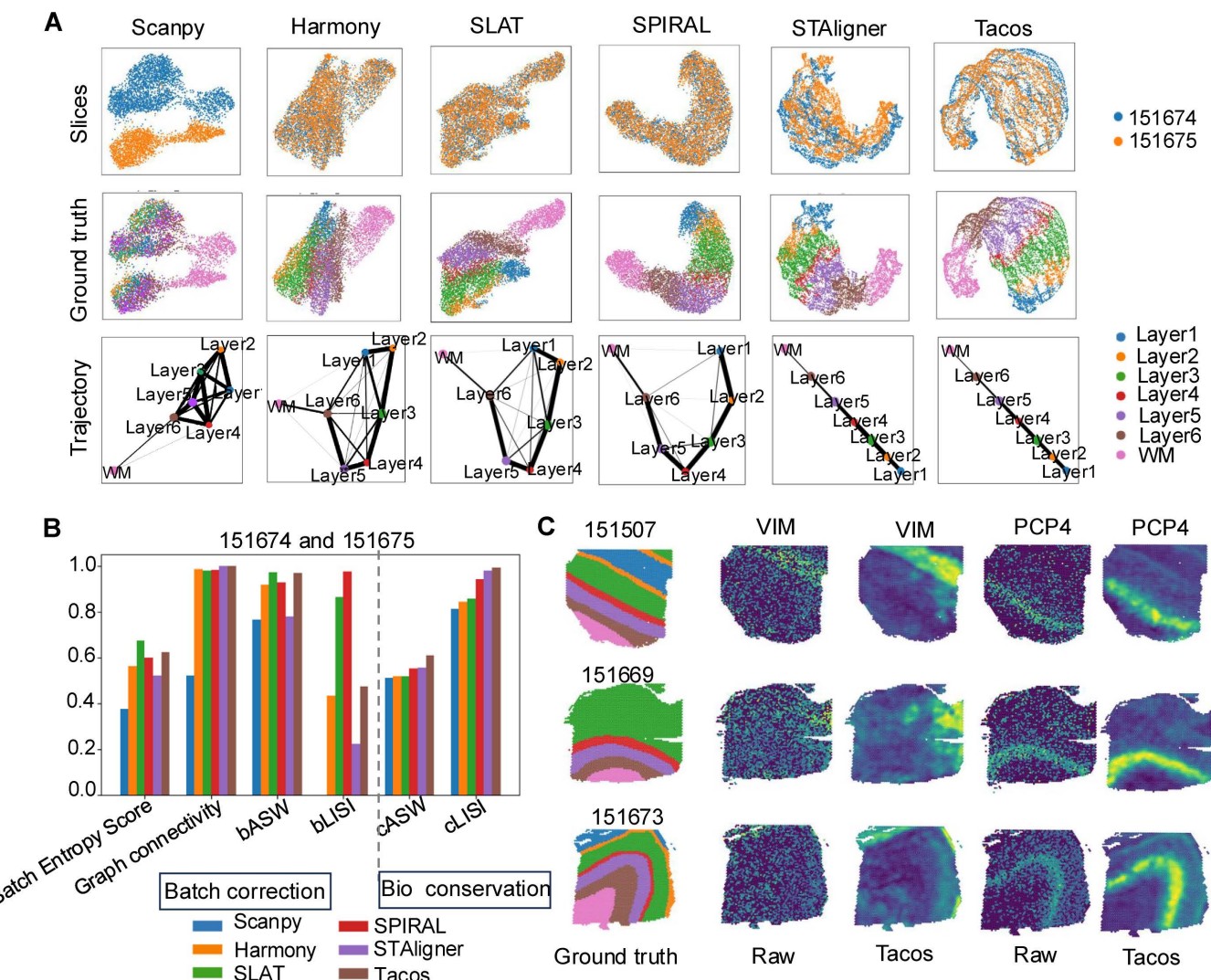

**Fig 2. Benchmarking Tacos with other methods on DLPFC datasets from 10X Visium.** (A) UMAP and PAGA visualization of aligned space of Scanpy, Harmony, SLAT, SPIRAL, STAligner and Tacos on slices 151674 and 151675. Spots are colored by slice numbers and annotated layers respectively. In a PAGA graph, thicker edges indicate stronger connections. (B) Bar plots of different metric scores of aligned performances of these methods on slices of 151674 and 151675. Batch Entropy Score, Graph connectivity, bASW and bLISI are metrics used for evaluating batch correction, while cASW and cLISI are used for evaluating biological conservation. (C) Visualization of the reconstructed marker gene (VIM for Layer1, PCP4 for Layer5) for slices 151507, 151669 and 151673, respectively.

layers. The PAGA graphs from STAligner and Tacos clearly depicted a linear developmental trajectory, whereas Scanpy, Harmony, SPIRAL and SLAT were unable to capture such pattern.

Then we applied Tacos to integrate non-adjacent slices (slice number: 151508 and 151675), which were derived from different donors. This setup introduced greater variability between the slices, making the task of alignment be difficult. The developmental trajectory inferred on the integrated embeddings of Tacos remained consistent in a linear developmental trajectory (Fig Aa in S1 Text). In comparison, STAligner was slightly less effective in maintaining the integrity of the hierarchical structure. On the other hand, Harmony, SLAT and SPIRAL exhibited increased misalignments, likely exacerbated by the inherent donor-specific differences between the slices. These misalignments disrupted the continuity of key structures, resulting

in less reliable representation of the underlying biological architecture. We employed batch effect removal metrics (Methods; Batch Entropy Score, Graph connectivity, bASW, bLISI) and label preserved metrics (cASW, cLISI) to quantitively compare these methods quantitively. Tacos consistently ranked in the top for capturing the true layers (Figs 2C, Ab in S1 Text). SLAT and SPIRAL demonstrated the best alignment performance in terms of batch entropy score. However, it had less scores in preserving annotated layers.

We compared the community-enhanced graph contrastive learning with graph convolutional network (GCN) encoder and graph contrastive learning (GraphCL) on slices 151508 and 151675 of DLPFC data. Specially, we replaced the community-enhanced graph contrastive learning with GCN and GraphCL, while maintaining other structures in Tacos. The community-enhanced graph contrastive learning approach consistently exhibited the best overall performance (Table A in S1 Text). We also applied PAGA to the embeddings of each method and found that the community-enhanced graph contrastive learning method could accurately capture the liner structures between different layers than that of GCN and GraphCL (Fig B in S1 Text).

Finally, we investigated the effectiveness of Tacos in data denoising using one slice from each donor (slice number: 151507, 151669 and 151673). Specifically, we focused on the marker genes associated with distinct brain layers, such as VIM for Layer 1 and PCP4 for Layer 5. Tacos accurately recovered the marker expression levels (Fig 2D). For example, the recovered spatial pattern of VIM was highly consistent with that of Layer 1 in slice 151673. Moreover, Tacos is able to integrate multiple slices at the same time. Specially, we applied Tacos to integrate adjacent four slices of each donor on DLPFC data. We found that Tacos could also accurately detect the linear structures of adjacent layers (Fig C in S1 Text).

## Tacos identifies slice-specific structures with clear boundaries across different platforms

We employed Tacos to analyze spatial transcriptomics data of mouse olfactory bulb collected from Slide-seqV2 [5] and Stereo-seq [7], which had different sequencing resolutions (Table 1). We compared Tacos with Scanpy, Harmony, SLAT, SPIRAL and STAligner in integrating results. Specially, we identified clusters by applying Louvain [26,27] algorithm on PCA space of normalized data and the low-dimensional embeddings derived from Harmony, SLAT, SPIRAL, STAligner and Tacos, respectively. Then we annotated each cluster according to the laminar structure in the DAPI-stained image [28] and Allen Mouse Brain Atlas annotation. Tacos notably detected clear layer boundaries within the mouse olfactory bulb (MOB) tissue (Fig 3A). These two slices had strong batch effects (Fig 3B). Tacos aligned clusters with the same structures together while preserved the uniqueness of Slide-seqV2 specific structures.

**Table 1. Summary of the spatial datasets.**

| Tissue | Species | Resolution | Technology | Cells/spots |
|---|---|---|---|---|
| Brain | human | 55 $\mu m$ | 10x Visium | 3400~4800 |
| Olfactory bulb | mouse | 10 $\mu m$ | Slide-seqV2 | 20,139 |
| Olfactory bulb | mouse | 220 $nm$ | Stereo-seq | 19,109 |
| Whole embryo | mouse | 220 nm | Stereo-seq | 5,913 |
| Whole embryo | mouse | subcellular | seqFISH | ~ 10, 000 |
| Hippocampus | mouse | 10 $\mu m$ | Slide-seqV2 | 12,000~15,100 |
| Brain | human | single cell | Xenium | 39427 |

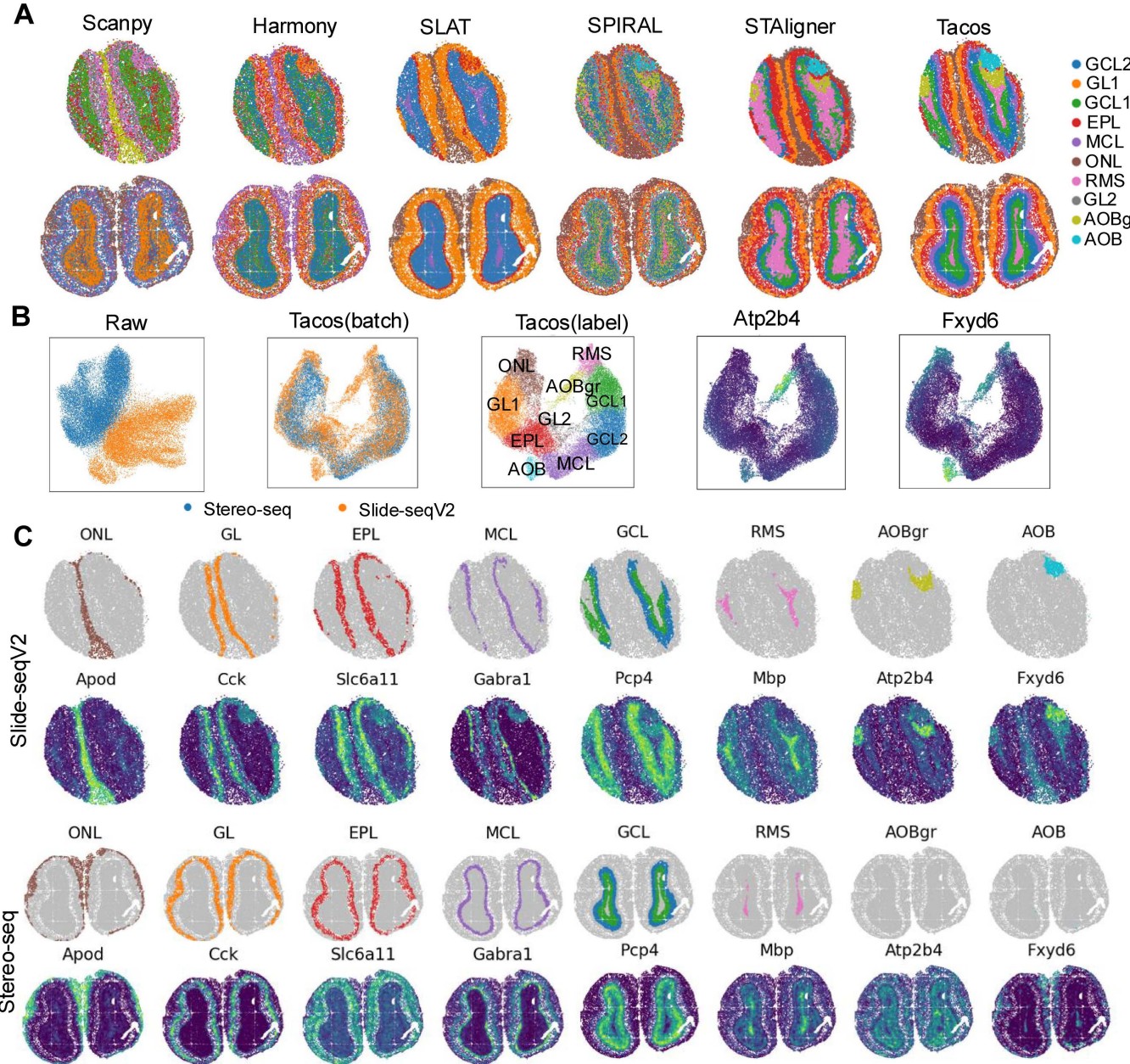

**Fig 3. Benchmarking Tacos with other methods on the MOB dataset.** (A) Visualization of spatial domain detection results of Tacos, STAligner, SPIRAL, SLAT, Harmony and Scanpy. (B) UMAP visualization of Tacos aligned space colored by louvain clusters and platforms. The expression levels of Atp2b4 and Fxyd6 are also shown. (C) Spatial visualization of spatial domains identified by Tacos and corresponding marker genes from Slide-seq slice (top) and Stereo-seq slice (bottom). The layers progressed from the outer to the inner layers, including the olfactory nerve layer (ONL), glomerular layer (GL), external plexiform layer (EPL), mitral cell layer (MCL), granule cell layer (GCL), rostral migratory stream (RMS), granular layer of the accessory olfactory bulb and accessory olfactory bulb.

Specially, the accessory olfactory bulb (AOB) and the granular layer of the accessory olfactory bulb (AOBgr) were Slide-seqV2 specific clusters, which were consistent with the spatially expressed patterns of their corresponding markers (Fxyd6 and Atp2b4). Tacos demonstrated the ability to preserve slice-specific structures while effectively integrating the shared laminar architecture, which avoided the risk of overcorrecting for batch effects.

To further investigate the shared clusters, we compared the spatial pattern of each cluster and its corresponding denoised marker gene. The shared clusters were spatially ordered from the outer to the inner layers of the MOB, including olfactory nerve layer (ONL), glomerular layer (GL), mitral cell layer (MCL), granule cell layer (GCL) and rostral migratory stream (RMS) (Fig 3C). Tacos accurately detected the RMS structure, a crucial structure in the mouse olfactory bulb slices. The boundaries of adjacent layers detected by STAligner and SPIRAL were not clear. Scanpy and Harmony failed to detect the structures, while SLAT only revealed the boundaries of major clusters.

## Tacos accurately maps tissues during different developmental stages of mouse embryo across different platforms

We employed Tacos to analyze spatial transcriptomic data of mouse embryos at E8.5 and E9.5 from seqFISH [29] and Stereo-seq [7], respectively. These two slices had different resolutions (Table 1). In addition to the technical variations, these two slices also exhibited differences in different developmental stages. We annotated each slice with the label from previous studies [7,29] (Fig 4A). Then we compared the performance of Tacos with Harmony, SLAT, SPIRAL and STAligner in removing batch effect and preserving cell types. Tacos, SPIRAL, SLAT and Tacos successfully aligned the slices despite their substantial technical differences, while Harmony and STAligner failed to remove the technological discrepancies (Fig 4B). SPIRAL, SLAT and Tacos had higher scores using batch effect removal-related metrics. However, SLAT and SPIRAL had less scores than Tacos using the cell type preserved-related metrics (Fig 4C). Tacos had the largest cASW scores, suggesting that Tacos outperformed other methods in preserving data structures.

Next, we investigated the aligned structures in the integrated space of Tacos. Specially, we examined pairs of spots with the same cell types across slices (Fig 4D). Tacos accurately aligned the neural crest from seqFISH and Stereo-seq. The neural crest displayed a dispersed pattern due to its complex migratory behavior and its role in differentiating into various cell types across multiple regions during embryonic development [30]. Tacos aligned Cardiomyocytes (seqFISH) and Heart (Stereo-seq) spots together. Moreover, Tacos accurately aligned the brain regions from Stereo-seq with the Forebrain/Midbrain/Hindbrain regions from seqFISH. Spots corresponding to the Aorta–gonad–mesonephros (AGM) region from Stereo-seq exhibited closely to the Splanchnic mesoderm and Intermediate mesoderm from seqFISH. The AGM is a critical site for hematopoiesis in mammals [31], and the mesonephros originates from the intermediate mesoderm [32]. Additionally, cells from the Splanchnic mesoderm serve as vasculogenic precursors, contributing to the formation of blood vessels and cells [33].

To further explore the biological relationships between tissues, we used a Sankey diagram based on Euclidean distances. Beyond the aforementioned pairs, we discovered significant connections between the Neural crest (Stereo-seq) and Cranial mesoderm (seqFISH), a relationship supported by previous studies, which highlighted the crucial interactions between these two regions during craniofacial development, facilitating coordination and regulatory processes [34]. Another strong association was observed between the Liver (Stereo-seq) and the Gut tube (seqFISH), which reflected their developmental connection, as the foregut, part of the gut tube, gave rise to the liver during organogenesis [35] (Fig 4E). Overall, Tacos accurately mapped tissues across different developmental stages, providing a comprehensive view of embryonic development trajectories.

## Tacos accurately detects different sophisticated structures between healthy and Alzheimer disease slices

Next, we applied Tacos to the spatial transcriptomics data of a healthy mouse hippocampal sample [5] and an Alzheimer's disease (AD) sample [36]. These two data were obtained by

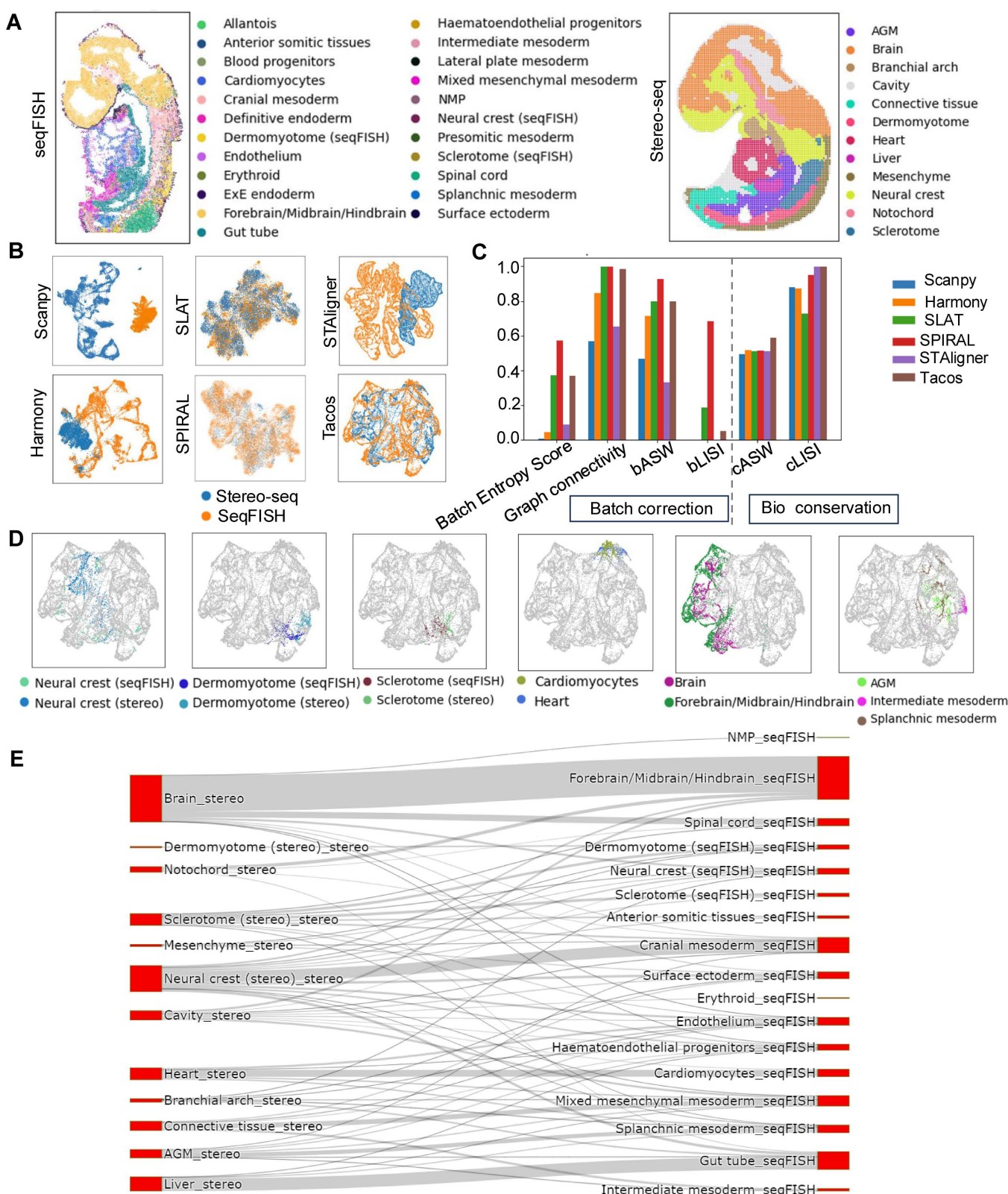

**Fig 4. The performance of Tacos on datasets of mouse embryo from seqFISH and Stereo-seq.** (A) Spatial visualization of mouse embryo from seqFISH and Stereo-seq with the annotated label. (B) UMAP visualization of aligned embeddings obtained by Scanpy, Harmony, SLAT, STAligner, SPIRAL and Tacos. (C) Bar plots of the metric scores of these methods in aligning mouse embryo slices. (D) UMAP visualization of aligned slices. Spots with the same labels are colored by different colors, while spots with different labels are colored as grey. (E) Sankey plot of connection between seqFISH and Stereo-seq on low embedding of Tacos.

Slide-seqV2 platform. We identified spatial domains by Louvain algorithm on the embeddings of Tacos and compared the results of Harmony, SLAT, SPIRAL and STAligner, respectively. Tacos and STAligner effectively delineated distinct anatomical structures within the hippocampus. In contrast, Harmony, SLAT and SPIRAL struggled to produce clear boundaries between spatial domains, which made it challenging to identify some of the more subtle structural variations (Fig 5A).

Precise boundary detection is crucial for identifying pathologically relevant regions. We found that Tacos was able to capture the well-characterized hippocampal formations, such as the cord-like Cornu Ammonis (CA1, CA2, and CA3) and the arrow-like dentate gyrus (DG), structures that were prominently recognizable and essential for hippocampal function (Fig 5B). While STAligner, could only discern the CA1 and CA3 regions, failed to detect the relatively small CA2 region, which was a transitional area that bridged CA1 and CA3. A previous study had shown that CA2 had distinct characteristics and treated it as an independent domain [37]. Moreover, Tacos efficiently denoised the spatial transcriptomic data and the recovered spatial patterns of marker genes were consistent with the spatial domains (Fig 5C). Aβ plaques, composed of aggregated Aβ protein polymers, are a hallmark of Alzheimer's disease and serve as one of the key pathological indicators [38]. These plaques were highly localized within the brain and are thought to contribute to neurodegeneration by disrupting neural function. We observed a strong spatial association between the early upregulation of C1q genes (C1qa, C1qb, C1qc)—which were implicated in the formation of Aβ plaques [36]—and their high expression specifically in regions dense with Aβ deposits (Fig 5D). Moreover, we detected a significant spatial correlation between the expression of Csf1r, a key growth factor

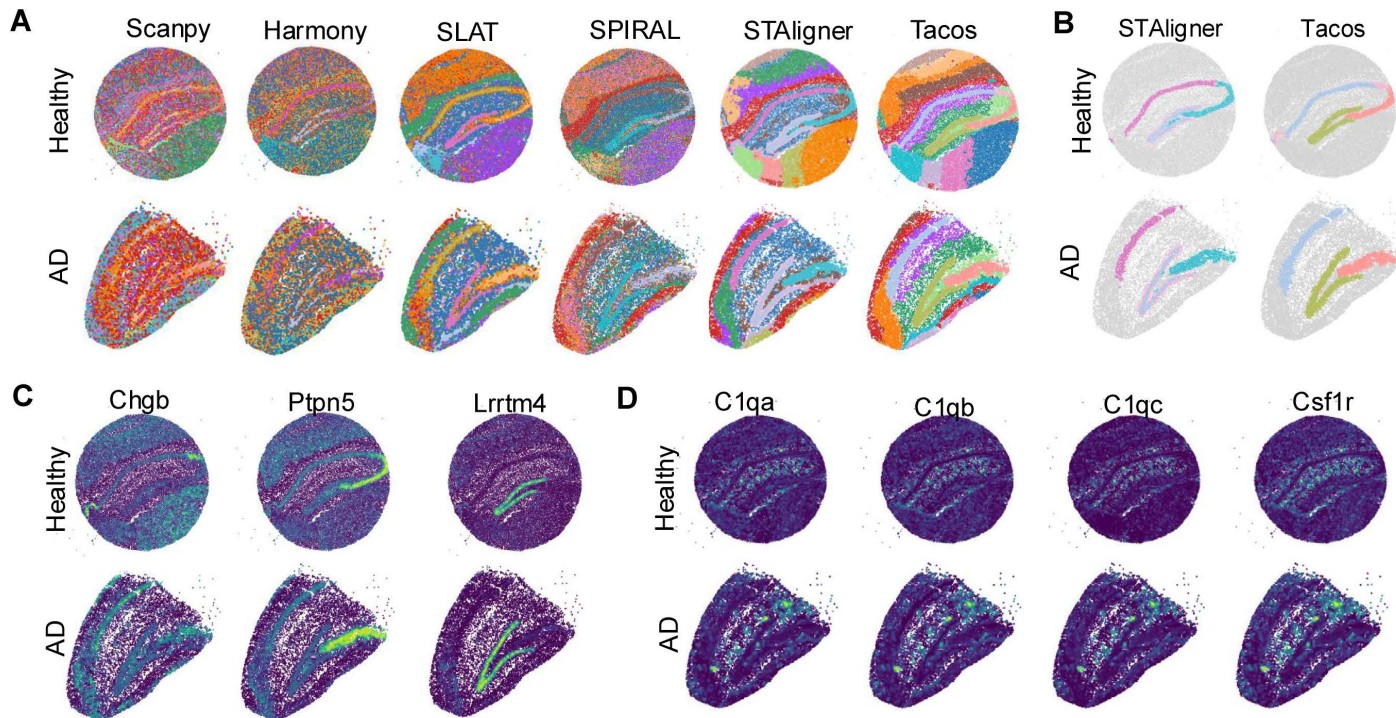

**Fig 5. The performance of Tacos on healthy hippocampus slice and Alzheimer hippocampus slice.** (A) Spatial visualization of domains detected by Scanpy, Harmony, SLAT, SPIRAL, STAligner and Tacos. (B) Spatial visualization of domains corresponding to CA1, CA2, CA3 and DG from Tacos and STAligner. (C) Spatial visualization of marker genes of spatial domain CA1, CA2, CA3 and DG. (D) Spatial visualization of disease relative genes in two slices including C1qa, C1qb, C1qc and Csf1r.

receptor involved in microglial function, and Aβ plaque regions [36]. Microglia, the brain's resident immune cells, are thought to play a critical role in the progression of AD, particularly in the clearance of Aβ deposits [36,39].

Finally, we applied Tacos to analyze the human healthy and AD slices coming from Xenium [40]. These two slices had large batch effects and Tacos aligned the healthy and Alzheimer's disease slices (Fig 6A). We further compared the domains 1, 3, and 6, which had regular spatial patterns in both healthy and Alzheimer's disease slices (Fig 6B). Interestingly, in the Alzheimer's disease slice, we observed that the expression level of CHODL was relatively higher in domain 1. CHODL has been implicated in influencing cell survival and neuron growth in animal models, suggesting that its elevated expression in this domain may reflect an adaptive [41] or pathological response in the Alzheimer's brain.

## Discussion

Aligning spatial transcriptomics coming from different platforms or experiments is a major challenge problem. In this study, we present a graph contrastive learning-based method Tacos to integrate spatial transcriptomic data from different conditions. Tacos considers the heterogeneity of spatial domains and adopts a community enhanced strategy to build augmented graph views used for learning embeddings of spatial transcriptomics data. We apply Tacos to many spatial transcriptomics datasets coming from different platforms with their resolutions varying from subcellular to multicellular levels. Comprehensive benchmark analyses with other methods show its superior performance. Compared to other aligned methods such as SLAT and STAligner, Tacos exhibits strong robustness in aligning multiple spatial transcriptomic data, especially for data coming from different sequencing platforms.

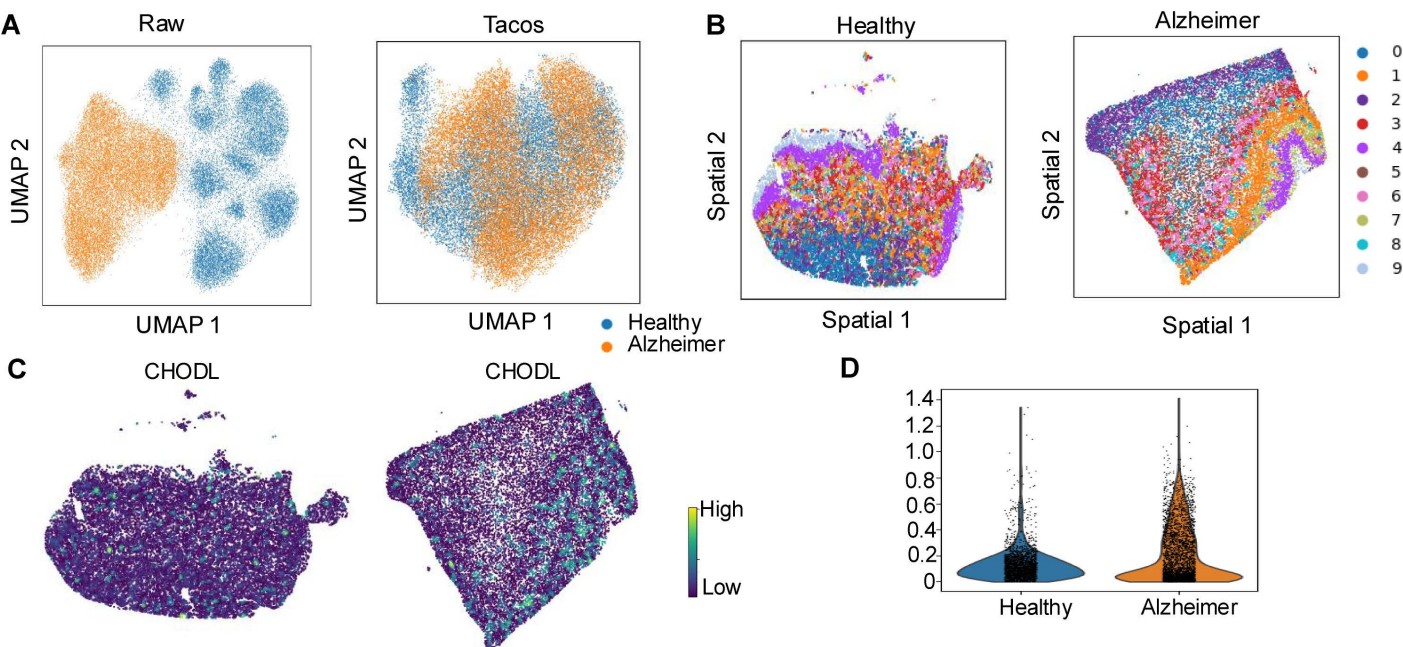

**Fig 6. The performance of Tacos on datasets of Xenium (Healthy and Alzheimer disease human brain).** (A) UMAP plot of raw data (left) and the aligned low-dimensional embedding of Tacos (right). (B) Spatially visualization of domains detected on healthy slice (left) and the Alzheimer disease slice (right). (C) The distribution of denoised CHODL by Tacos on the spatial location. (D) Violin plot of CHODL in cluster 1 across healthy and Alzheimer slices.

Though Tacos achieves good performance in integrating spatial transcriptomics data, its performance can be further improved with the increasing numbers of annotated slices. Tacos is an unsupervised method, which does not take advantage of the annotated information. A semi-supervised method will be needed in the future. Moreover, the current framework of Tacos focuses on and transcriptomics information and spatial coordinates without considering histological images. Future work is expected to incorporate such information to facilitate more comprehensive understand of spatial heterogeneity. Large memory utilization is a critical issue for graph-based methods. To reduce memory usage, we have optimized our code by modifying the training process. Instead of feeding all slices into the model simultaneously, we process them individually during training, which can decrease memory requirements. Additionally, a lightweight graph convolutional approach in the community-enhanced graph learning step will be explored in the future.

## Methods

### Datasets and data preprocessing

In this study, we applied Tacos to six ST datasets from different platforms, including 10x Visium, Stereo-seq, Slide-seqV2, seqFISH, Xenium and MERFISH (Table 1). The dorsolateral prefrontal cortex (DLPFC) dataset was from 3 donors, and each donor had 4 slices (labeled as 151507–151510, 151669–151672 and 151673–151676). Each slice was manually annotated into 4–6 cortical layers and white matter [7]. The mouse olfactory bulb dataset included two slices obtained from Slide-seqV2 [5] and Stereo-seq [7]. The mouse embryo dataset consisted of two slices of somite stage E8.5 and stage E9.5 obtained from seqFISH [42] and Stereo-seq [7]. The mouse hippocampus dataset contained one healthy slice and one Alzheimer diseased slice coming from Slide-seqV2 platform. There are 2 slices in the high-resolution hippocampus dataset coming from Xenium [40].

We used scanpy [23] package to preprocess these data. Specially, the gene expression values were normalized by dividing the total UMI count across all genes in each spot and multiplying 10,000. And then the data was transformed to a natural log scale with pseud-count equaling 1. In this study, we used all genes if the spatial transcriptomics data has fewer than 3000 genes. Otherwise, we selected high variable genes using the function sc.pp.highly_variable_genes with flavor="seurat_v3" in Scanpy package and top 3000 high variable genes were selected. Finally, genes in the intersection set of high variable genes of all slices were used.

### Building spatial graph for spatial transcriptomic data

We constructed spatial graph $G_t$ for slice $t$ using alpha complex [43] based on the spatial coordinate $Y_t \in R^{n \times 2}$. Spots were vertexes and the normalized expression matrix $X_t \in R^{n \times m}$ was treated as feature. Then we built geometry-aware spatial proximity adjacency matrix $A_t \in R^{n \times n}$ for the graph $G_t$. Specially, we computed a unique site $V(s)$ for each spot or cell $s$ using $V(s) = \{\|x - s\| \le \|x - v\|, \forall v \Subset C\}$, where $C$ was the set of coordinates for all spots. $V(s)$ is a spot set composed of any point $x$ which is closer to $s$ than to any other point $v$. Next, we identified the neighborhood edges $E$ by connecting spots $i$ and $j$ as follows:

$$E = \left\{ (i, j) \mid \cap_{s \in \{i, j\}} \left( V(s) \right) \cap R(s, r) \right\}, \tag{1}$$

where $R(s, r)$ is a local neighborhood around $s$ with a radius $r$. The radius $r$ was estimated by the mean distance of $k$ nearest neighbors of the spot based on spatial coordinates. The alpha complex-based method uses geometric structures to define neighborhoods, which makes it more suitable for spatial data with non-uniform densities.

## Extracting low-dimensional embeddings with community-enhanced encoder

We used a contrastive learning strategy to capture local information on spatial neighbor graph $G_t$. In the first stage, we detected the putative communities using Leiden [44] on the normalized gene expression or on the embedding obtained by the following strategy. Specially, we generated two augmented graph views $G_t^1$ and $G_t^2$ by randomly perturbated the graph $G_t$. Then we adopted a two-layer Graph Convolutional Network (GCN) on $G_t^1$ and $G_t^2$ to extract low-dimensional embeddings $Z_t^1$ and $Z_t^2$. For the gene expression data with $n$ genes, the first layer transforms the input feature dimension from $n$ to $2h$ followed with Parametric Rectified Linear Unit (PReLU) activation, while the second layer reduces the feature dimension to $h$ with the default value equaling 50. We used InfoNCE loss [45] to train the GCN as follows:

$$\mathcal{L}_t = -\frac{1}{n}\sum_{i=1}^{n} log \frac{\exp\left(s_{ii}^{(1,2)}\right)}{\sum_{j=1,j\neq i}^{n} \exp\left(s_{ij}^{(1,1)}\right) + \sum_{j=1}^{n} \exp\left(s_{ij}^{(1,2)}\right)}, \tag{2}$$

where $s_{ij}^{(1,2)} = sim\left(Z_{i:}^1, Z_{j:}^2\right)/\tau$ and $\tau$ is the temperature of similarity. The tissue usually had complex spatial structures and the slices coming from different platforms might be in different resolutions, leading to the heterogenous communities within the spatial graphs. Therefore, we extracted the low-dimensional embeddings of slices by a community-enhanced encoder in the second stage [46]. The number of edges in community $c$ was represented by $\left|\mathcal{E}_c\right|$, and the number of edges in the whole graph was represented by $\left|\mathcal{E}\right|$. We defined the community strength of $c$ by $\mathcal{S}_c$ as follows:

$$\mathcal{S}_c = \frac{\left|\mathcal{E}_c\right|}{\left|\mathcal{E}\right|} - \frac{(\sum_{v\in c} d(v))^2}{4\left|\mathcal{E}\right|^2} \tag{3}$$

where $d(\cdot)$ was node degree. The community strength matrix was represented by $S$. Two strategies (communal attribute voting and communal edge dropping) were used to generate augmented graph views [46]. The dimensional attributes with higher scores were more likely to be masked. The probability of attribute masking $w_t$ was computed as: $w_t = Norm\left(abs\left(X_t\right)^T H\mathcal{S}\right)$, where $Norm(\cdot)$ was one-dimensional normalization operation and $H \in \{0,1\}^{n_t \times |C|}$ was community indicator. Then the masking matrices were generated from Bernoulli distributions. Afterward, the masking matrices were multiplied by the data matrices, achieving the purpose of masking a portion of the attribute as follows:

$$m_t^1 \sim Bernoulli\left(1 - w_t p_t^1\right), \tag{4}$$

$$m_t^2 \sim Bernoulli\left(1 - w_t p_t^2\right), \tag{5}$$

$$X_t^1 = m_t^1 \circ X_t, \tag{6}$$

$$X_t^2 = m_t^2 \circ X_t, \tag{7}$$

where $\circ$ was Hadamard product, $p_t^1$ and $p_t^2$ were two hyperparameters.

For each edge $e = \left(u_i, u_j\right)$, the probability of masking was computed by:

$$w(e) = \begin{cases} H_{i:}S, (HH^T \circ A)_{i,j} = 1 \\ -\left(H_{i:}\mathcal{S} + H_{j:}\mathcal{S}\right), otherwise \end{cases}, \tag{8}$$

$$w_e = Norm(w(e)), \tag{9}$$

where $\left(HH^T \circ A\right)_{i,j} = 1$ meant $\left(u_i, u_j\right)$ was an intra-community edge. The edge masking results were:

$$m_e^1 \sim Bernoulli\left(w_e p_e^1\right), \tag{10}$$

$$m_e^2 \sim Bernoulli\left(w_e p_e^2\right), \tag{11}$$

where $p_e^1$ and $p_e^2$ were two hyperparameters. To mask the graph edge, we manipulated the adjacency matrix of the graph:

$$A_e^1 = \left[m_{e,(i,j)}^1 A_{t,(i,j)}\right], \tag{12}$$

$$A_e^2 = \left[m_{e,(i,j)}^2 A_{t,(i,j)}\right], \tag{13}$$

By this way, we obtained two augmented graph views containing community strength information. Then we employed a two-layer GCN to learn the low-dimensional embeddings of these two graph views by:

$$Z_t^1 = Encoder\left(X_t^1, A_e^1\right), \tag{14}$$

$$Z_t^2 = Encoder\left(X_t^2, A_e^2\right), \tag{15}$$

We used InfoNCE loss to train the GCN as follows:

$$\mathcal{L}_t = -\frac{1}{n}\sum_{i=1}^{n} log \frac{\exp\left(\tilde{s}_{ii}^{(1,2)}\right)}{\sum_{j=1, j\neq i}^{n} \exp\left(\tilde{s}_{ij}^{(1,1)}\right) + \sum_{j=1}^{n} \exp\left(\tilde{s}_{ij}^{(1,2)}\right)}, \tag{16}$$

$$\tilde{s}_{ij}^{(1,2)} = s_{ij}^{(1,2)} + \gamma(k)\left(H_{i:} + H_{j:}\right)S, \tag{17}$$

where $s_{ij}^{(1,2)} = sim\left(Z_{i:}^1, Z_{j:}^2\right)/\tau$. $sim\left(z_i, z_j\right) = \frac{z_i^T z_j}{\|z_i\|\|z_j\|}$. $\gamma(k)$ was a dynamic balancing coefficient that could gradually increase during training $\gamma(k; k_0, \gamma_{max}) = min\{max\{0, k - k_0\}, \gamma_{max}\}$. Finally, the embedding $Z_t$ obtained from the GCN trained in the second stage on graph $G_t$.

## Spatial similarity constraint within each slice

To capture the similarity between the spatially neighbored spots within each slice, we added a spatial similarity constraint [14] on the low-dimensional embedding $Z_t$ of $t$-th slice as follows:

$$L^t_{within} = \sum_{i=1}^{n_t} \sum_{j=1}^{n_t} \frac{D^{(s)}_{i,j} * \left(1 - D^{(Z_t)}_{i,j}\right)}{n_t * n_t}, \tag{18}$$

where $D^{(s)}_{i,j}$ represented the spatial Euclidean distance between spots $i$ and spots $j$, and $D^{(Z_t)}_{i,j}$ represented the Euclidean distance between low-dimensional embeddings of spot $i$ and spot $j$.

## Aligned constraint across different slices

We adopted the triplet learning to align different slices, which had shown superior performance in removing batch effects in scRNA-seq and spatial transcriptomics data integration [20,47]. We identified MNN pairs on the low-dimensional embeddings of these slices and then treated the MNN pairs as anchor and positive points. Negative points were randomly sampled. The aligned constraint minimizes the distance between positive pairs and maximizes the distance between negative pairs as follows:

$$L^{t_1 t_2}_{across} = \frac{1}{N_{t_1 t_2}} \times \sum_{N_{t_1 t_2}}^{(a,p,n) \in C_{t_1 t_2}} \max\left( \| Z^{t_1}_{a:} - Z^{t_2}_{p:} \|_2 - \| Z^{t_1}_{a:} - Z^{t_1}_{n:} \|_2 + \theta, 0 \right), \tag{19}$$

where $Z^{t_1}_{a:}$, $Z^{t_2}_{p:}$, and $Z^{t_1}_{n:}$ represent the anchor point, positive point, and negative point, respectively. $C_{t_1 t_2}$ denotes the set of triplets and $N_{t_1 t_2}$ is the number of $C_{t_1 t_2}$. $\theta$ (default: 1.0) is the margin parameter.

Finally, the total loss function is:

$$L = \sum_t \mathcal{L}_t + \alpha \sum_t \mathcal{L}^t_{within} + \beta \sum_{t_1} \sum_{t_1 < t_2} \mathcal{L}^{t_1 t_2}_{across}, \tag{20}$$

where $\alpha$ and $\beta$ are hyperparameters that control the relative weighting of different components of the loss. We adopted the Adam optimizer in PyTorch with a learning rate being 0.001 and epoch equaling 2000 on a NVIDIA RTX 3090 GPU. The computational efficiency of Tacos on each dataset were shown in Table B in S1 Text.

## Identification of spatial domains

We identified spatial domains with mclust or community-graph based Louvain or Leiden algorithm on the obtained embeddings of Tacos and other methods. Specially, we recommend to use mclust for data from the 10x Visium platform, while we suggest to use Leiden for other platforms.

## Spatial transcriptomics data denoising

The learnt embeddings preserve the local context for each spot, which have potential in denoising the spatial transcriptomics data. Specially, we fed the embeddings into an extra two-layer perceptron neural network, which had a symmetric structure to the GCN encoder that map the low-dimensional embedding back to the original feature space. The corresponding loss function is:

$$\mathcal{L}_{recon} = \sum_t f\left(Z_t\right) - X_{tt}, \tag{21}$$

where $X_t$ is the normalized gene expression matrix of slice $t$.

## Evaluation metrics

We used the following metrics to evaluate the performance of integrating spatial domains of different slices.

## Batch entropy score

Batch entropy score [48] quantifies the degree of alignment and it can be calculated as:

$$E = \sum_k^{i=1} \frac{\frac{p_i}{P_i}}{\frac{\sum_{j=1}^{k} p_j}{P_j}} log\left(\frac{\frac{p_i}{P_i}}{\frac{\sum_{j=1}^{k} p_j}{P_j}}\right), \tag{22}$$

where $p_i$ is the proportion of spots from the $i$-th slice in a given region, $P_i$ is the total number of spots of the $i$-th slice. A higher batch entropy score indicates better alignment of different slices.

## Label ASW and batch ASW score

Average silhouette width (ASW [49]) quantifies the degree of separation among spots with different labels. Higher label ASW score indicates better performance. Label ASW score is calculated as $S_i = (b-a)/max(b,a)$, where $b$ presents the average distance of a slice $i$ with other slices from the same cluster and $a$ is the average distance of slice $i$ with other slices from different clusters. To ensure higher scores indicate better slice mixing, batch ASW score (bASW) is scaled by subtracting them from 1. Label ASW score (cASW) and batch ASW are calculated on PCA embedding.

## cLISI and bLISI

The inverse Simpson's index (LISI) [11] reports the effective indicating perfect separation labels and perfect mixing slices. Suppose there are $B$ slices, we rescaled cLISI and bLISI to the range 0–1. Specially, $cLISI = median\left(\frac{B-x}{B-1}\right)$, $bLISI = median\left(\frac{x-1}{B-1}\right)$, where x is LISI score.

## Graph connectivity [50]

Graph connectivity (GC) metric measures whether the kNN graph representation of the integrated data connects cells with the same label. For a subset kNN graph $G_c = (N_c, E_c)$ which only contains cells with label $c$, GC is calculated as:

$$GC = \frac{1}{|C|} \sum_{c \in C} \frac{\left|LCC(G_c)\right|}{|N_c|}, \tag{23}$$

where $\left|LCC(*)\right|$ is the number of nodes in the largest connected component of the graph.

## Supporting information

**S1 Text. Supplementary materials.**
(PDF)

## Author contributions

**Conceptualization:** Lihua Zhang.

**Investigation:** Lihua Zhang.

**Methodology:** Wenqian Tu.

**Project administration:** Lihua Zhang.

**Software:** Wenqian Tu.

**Supervision:** Lihua Zhang.

**Validation:** Wenqian Tu.

**Visualization:** Wenqian Tu.

**Writing – original draft:** Wenqian Tu, Lihua Zhang.

**Writing – review & editing:** Wenqian Tu, Lihua Zhang.

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
