## [Decision Letter · Decision Letter 0]

14 Jan 2025

PCOMPBIOL-D-24-01959

Integrating multiple spatial transcriptomics data using community-enhanced graph contrastive learning

PLOS Computational Biology

Dear Dr. Zhang,

Thank you for submitting your manuscript to PLOS Computational Biology. After careful consideration, we feel that it has merit but does not fully meet PLOS Computational Biology's publication criteria as it currently stands. Therefore, we invite you to submit a revised version of the manuscript that addresses the points raised during the review process.

Please submit your revised manuscript within 60 days Mar 16 2025 11:59PM. If you will need more time than this to complete your revisions, please reply to this message or contact the journal office at ploscompbiol@plos.org. Please include the following items when submitting your revised manuscript:

We look forward to receiving your revised manuscript.

Kind regards,

Wei Li, Ph.D.

Academic Editor

PLOS Computational Biology

Jian Ma

Section Editor

PLOS Computational Biology

**Journal Requirements:**

4) Please amend your detailed Financial Disclosure statement. This is published with the article. It must therefore be completed in full sentences and contain the exact wording you wish to be published. Please ensure that the funders and grant numbers match between the Financial Disclosure field and the Funding Information tab in your submission form. Note that the funders must be provided in the same order in both places as well.

**Reviewers' comments:**

Reviewer's Responses to Questions

**Comments to the Authors:**

Reviewer #1: Tu et al., in their research article titled “Integrating multiple spatial transcriptomics data using community-enhanced graph contrastive learning,” showcase the use of graph learning method to integrate multiple spatial transcriptomics data from different platforms with different scenarios. The method development contributes to easing the complexities posed by spatial transcriptome data, especially the batch effect and noise impact. At its current stage, the article addresses technical defects of spatial transcriptomics rather than providing a certain degree of biological insights and aspects. The tool presented in the article is limited to integrating datasets from the same or different platforms, making the article in its present form unsuitable for publication.

However, the article could be significantly improved if the authors added points as discussed below:

1) In the Overview of Tacos sub-section of the Result section, the author showcases a figure representing the schema of the graph theory algorithm. However, it is not mentioned why graph contrastive learning is the better choice. What were the criteria for selecting this algorithm? A better alternative will be to train different algorithms and finalize the algorithm that is best suited for the dataset. This validates and gives a better rationale for the tool development

2) The author shows the smoothness of integration but fails to justify how non-computational biologists or bioinformaticians could utilize the present tool. Could the tool in its present form be integrated into well-known spatial transcriptomics analysis? If so, how could they integrate it?

3) The tool's computational efficiency is not extensively discussed. Moreover, if the tool has extensive memory utilization, a section or supplementary note should discuss how multithreading or multiprocessing could elevate process execution.

4) Both scRNA-seq and spatial transcriptomics are used to find novel biological markers or insights. However, the present tool does not provide a better alternative to the existing tool.

4) The GitHub page appears to be well maintained; however, the readme and license file need to be updated

Reviewer #2: The manuscript introduces Tacos, a novel community-enhanced graph contrastive learning-based method designed to integrate multiple spatial transcriptomics datasets across varying platforms and conditions. The method offers a promising tool for integrating spatial transcriptomics data, with potential applications in studying tissue architecture and disease progression. However several areas could benefit from further clarification to improve its overall impact and scientific rigor.

1. Given the align output is in a low-dimensional embedding space. Besides to the comparison to the Scanpy integration without batch effect removal, it is preferable to also compare it with Scanpy's batch-corrected integration methods like Harmony. Additionally, SPIRAL has the same design for spatial data integration across various experiments and technologies, SPIRAL also test the performance on DLPFC and MOA datasets. it would be beneficial to include SPIRAL into the comparisons as well.

2. How spatial gene expression data is denoised? A detailed explanation of the approach used to denoise spatial gene expression data are required.

3. Compared to ground truth annotations, Tacos had less scores in preserving annotated layers, Could you clarify how these scores are defined? Additionally, does this imply that Tacos may not effectively preserve the biological structures represented by the annotated layers?

4. The trajectory tree analysis appears unconventional. Could you clarify how the hierarchical organization of the dorsolateral prefrontal cortex tissue was interpreted?

5. All the test datasets in the manuscript appear to involve two-sample integration. It would be valuable to assess the performance of Tacos in integrating multiple samples simultaneously.

6. Although spatial domain is not the direct output of Tacos, it is important to provide a detailed description of how spatial domains were determined. Additionally, clarify the criteria and methods used to compare spatial domains across different approaches.

Reviewer #3: In this study, the authors present TACOS, an advanced method for spatial transcriptomics integration that combines graph convolutional neural networks (GCNs) with innovative data preprocessing and augmentation strategies. By constructing cell graphs that integrate spatial and gene expression information, the method provides a robust framework for capturing the intricate relationships within spatial transcriptomics datasets. Furthermore, TACOS leverages contrastive learning to enhance feature representations, enabling the alignment of spatial datasets with unprecedented accuracy. This approach not only achieves state-of-the-art performance in spatial data integration but also addresses key challenges in preserving spatial context and biological relevance. Importantly, the authors have developed TACOS as a Python package, ensuring its accessibility and utility for the research community. This work represents a significant advancement in the field of spatial transcriptomics, offering both methodological innovations and practical tools for researchers. While I recommend the paper for publication, addressing the following relatively minor comments will improve the manuscript's clarity and presentation.

1. Figure 1 and Caption: The figure does not provide adequate details to explain the model's training process. Specifically, the manuscript does not clarify how the deep learning model generates the aligned dataset or what specific steps are taken to achieve this alignment. Moreover, the caption does not describe the training objectives, such as the loss functions used in the model, or explain their significance in aligning the datasets. Adding these details would greatly enhance the clarity and accessibility of this figure, ensuring readers fully understand the underlying methodology.

2. Methods Section – "Datasets and Data Preprocessing": There is a typographical error in the term "pseud-count," which should be corrected to "pseudocount." Additionally, the manuscript does not adequately discuss the choice of the number of highly variable genes, a parameter that is crucial for the analysis. This choice can significantly affect the results, and explicitly presenting the criteria used to select this parameter is necessary for reproducibility and transparency. A brief justification for the chosen number would strengthen the methodological rigor of the study.

3. Methods Section – Graph Construction and Embeddings: The subsections "Building Spatial Graph for Spatial Transcriptomic Data" and "Extracting Low-Dimensional Embeddings with Community-Enhanced Encoder" require additional clarity in explaining the equations and symbols introduced. For example, the meaning of symbols such as "s" and "x" in the "1-skeleton" equation is not explained, leaving their interpretation ambiguous. Similarly, the rationale for using "V(s)" instead of directly constructing a k-nearest neighbor (KNN) graph based on spatial distances is not adequately articulated. Providing a clear explanation of how these techniques contribute to solving the spatial integration task would strengthen the section. Additionally, numbering the equations for easier reference and avoiding redundancy in their presentation would improve the overall organization and readability of the manuscript.

4. Model Architecture and Training Details: The manuscript omits critical information about the structure and optimization of the deep learning model. Details such as the architecture of each module during training, including the number of neurons, are not provided. Similarly, essential optimization parameters, such as the choice of optimizer, learning rate, and the number of training epochs, are missing. Furthermore, there is no discussion of runtime or computational efficiency, particularly concerning graph construction and graph convolutional operations. Since the method involves graph-based spot-level interactions, it is important to clarify whether full-batch mode is used for graph convolutions and, if so, discuss its implications for runtime and memory requirements. Including these details would provide a more comprehensive understanding of the computational aspects of the method.

5. Figure 2 and Baseline Comparisons: The use of Leiden clustering on embeddings derived from PCA in Scanpy is not appropriate. Leiden clustering is primarily designed for identifying cell types based on gene expression, not for analyzing spatial slices, making its relevance to this context questionable. Additionally, the comparison with the ground truth trajectory is unclear. The manuscript mentions performing PAGA using ground truth labels, but this approach does not seem to provide a fair baseline for evaluating spatial integration tasks. A more suitable baseline comparison that directly addresses the spatial integration objectives of the study would better validate the effectiveness of TACOS and bolster the credibility of the results.

**Have the authors made all data and (if applicable) computational code underlying the findings in their manuscript fully available?**

Reviewer #1: Yes

Reviewer #2: Yes

Reviewer #3: Yes

PLOS authors have the option to publish the peer review history of their article (what does this mean? ). If published, this will include your full peer review and any attached files.

**Do you want your identity to be public for this peer review?** For information about this choice, including consent withdrawal, please see our Privacy Policy .

Reviewer #1: No

Reviewer #2: No

Reviewer #3: No

**Figure resubmission:**
---

## [Decision Letter · Decision Letter 1]

10 Mar 2025

Dear Dr Zhang,

We are pleased to inform you that your manuscript 'Integrating multiple spatial transcriptomics data using community-enhanced graph contrastive learning' has been provisionally accepted for publication in PLOS Computational Biology.

Best regards,

Wei Li, Ph.D.

Academic Editor

PLOS Computational Biology

Jian Ma

Section Editor

PLOS Computational Biology

Reviewer's Responses to Questions

**Comments to the Authors:**

Reviewer #1: The author have addressed my concerns

Reviewer #2: The author has addressed all of my comments.

Reviewer #3: I appreciate the authors’ thorough and thoughtful responses to my comments. Their clarifications and additional analyses have effectively addressed my concerns, and I am satisfied with their revisions. I also commend the authors for their rigorous approach and the high quality of their work. This study represents an exciting advancement in the field, and I look forward to seeing its impact on the community.

**Have the authors made all data and (if applicable) computational code underlying the findings in their manuscript fully available?**

Reviewer #1: Yes

Reviewer #2: Yes

Reviewer #3: Yes

PLOS authors have the option to publish the peer review history of their article (what does this mean? ). If published, this will include your full peer review and any attached files.

**Do you want your identity to be public for this peer review?** For information about this choice, including consent withdrawal, please see our Privacy Policy .

Reviewer #1: No

Reviewer #2: No

Reviewer #3: No

---

## [Editor Report · Acceptance letter]

PCOMPBIOL-D-24-01959R1

Integrating multiple spatial transcriptomics data using community-enhanced graph contrastive learning

Dear Dr Zhang,

I am pleased to inform you that your manuscript has been formally accepted for publication in PLOS Computational Biology. Your manuscript is now with our production department and you will be notified of the publication date in due course.

With kind regards,

Zsofia Freund
